# Impact of Co-Occurrence of Obesity and SARS-CoV-2 Infection during Pregnancy on Placental Pathologies and Adverse Birth Outcomes: A Systematic Review and Narrative Synthesis

**DOI:** 10.3390/pathogens12040524

**Published:** 2023-03-27

**Authors:** Thaina Ferraz, Samantha J. Benton, Israa Zareef, Oluwatomike Aribaloye, Enrrico Bloise, Kristin L. Connor

**Affiliations:** 1Health Sciences, Carleton University, Ottawa, ON K1S 5B6, Canada; 2Department of Morphology, Federal University of Minas Gerais, Belo Horizonte 31270-901, Brazil

**Keywords:** maternal, obesity, SARS-CoV-2, placenta, pathology, neonatal, anthropometry

## Abstract

Obesity is a risk factor for severe COVID-19 disease during pregnancy. We hypothesized that the co-occurrence of high maternal body mass index (BMI) and gestational SARS-CoV-2 infection are detrimental to fetoplacental development. We conducted a systematic review following PRISMA/SWiM guidelines and 13 studies were eligible. In the case series studies (*n* = 7), the most frequent placental lesions reported in SARS-CoV-2(+) pregnancies with high maternal BMI were chronic inflammation (71.4%, 5/7 studies), fetal vascular malperfusion (FVM) (71.4%, 5/7 studies), maternal vascular malperfusion (MVM) (85.7%, 6/7 studies) and fibrinoids (100%, 7/7 studies). In the cohort studies (*n* = 4), three studies reported higher rates of chronic inflammation, MVM, FVM and fibrinoids in SARS-CoV-2(+) pregnancies with high maternal BMI (72%, *n* = 107/149; mean BMI of 30 kg/m^2^) compared to SARS-CoV-2(−) pregnancies with high BMI (7.4%, *n* = 10/135). In the fourth cohort study, common lesions observed in placentae from SARS-CoV-2(+) pregnancies with high BMI (*n* = 187 pregnancies; mean BMI of 30 kg/m^2^) were chronic inflammation (99%, 186/187), MVM (40%, *n* = 74/187) and FVM (26%, *n* = 48/187). BMI and SARS-CoV-2 infection had no effect on birth anthropometry. SARS-CoV-2 infection during pregnancy associates with increased prevalence of placental pathologies, and high BMI in these pregnancies could further affect fetoplacental trajectories.

## 1. Introduction 

Globally, the prevalence of overweight and obesity tripled between 1975 and 2020 [1]. Obesity represents a major public health concern as individuals with overweight or obesity are at increased risk of cardiovascular disease, diabetes, and death [2]. Of particular concern is the growing number of individuals entering into pregnancy with overweight or obesity [1]. In the United States, an estimated 31.1% of women of reproductive age (18–44 years) have obesity [3,4]. In Canada, between 2004 and 2014, 22% of pregnancies were complicated by maternal obesity and 24% by maternal overweight [5]. Further, the prevalence of overweight and obesity is rising in low- and middle-income countries (LMIC) [6]. For example, in India, the rate of overweight among reproductive aged women increased from 10.6% to 14.8% over a 10-year period [7]. Obesity in pregnancy is associated with an increased risk of comorbidities such as gestational hypertension, gestational diabetes and delivery of large for gestational age (LGA) babies compared to pregnancies without obesity [8]. In LMICs, the burden of adverse outcomes for pregnancies with obesity is exacerbated by limited access to reliable antenatal and postnatal care and interventions, leading to an increased risk of poor outcomes for mother and offspring [9].

During pregnancy, the placenta is the key barrier and conduit between the mother and fetus. Moreover, the placenta adjusts to the pregnancy environment in order to sustain fetal growth and development [10]. Proper placental development is critical for the adequate exchange of oxygen and nutrients between mother and fetus [11]. Abnormal placental development and function is associated with preterm delivery, fetal growth restriction (FGR) and preeclampsia [11]. Although not completely understood, maternal morbidities such as obesity may contribute to poor placental development and contribute to the pathogenesis of these pregnancy complications [12]. Obesity is known to induce systemic, low-grade chronic inflammation through the production and release of cytokines from adipose tissue [12,13]. During pregnancy, cytokines play an important role in mediating implantation [14], placentation [15] and parturition [16]. An exacerbated *in utero* inflammatory environment could lead to suboptimal placental development by impeding trophoblast cell function, the fetal-derived cells of the placenta. Obesity-driven factors such as cytokines, lipids, reactive oxygen species and hormones have been shown to impact migration and stimulate apoptosis in extravillous trophoblasts [17]. Leptin, an important cytokine produced by adipose tissue, can regulate placental angiogenesis, protein synthesis, and growth [18,19]. Increased levels of leptin, as commonly occurs in obesity, may lead to placental leptin resistance and decreased amino acid transport activity, thereby reducing nutrient availability required for fetal growth [20]. Additionally, evidence of altered placental development and function in obese pregnancies is shown by the increased prevalence of pathological lesions in the placenta. Commonly, inflammatory lesions [21] and maternal vascular lesions are observed in placentae from pregnancies with maternal obesity [22]. These lesions are associated with altered placental function and adverse outcomes such as fetal demise and FGR [23]. 

In addition to maternal obesity, other insults such as antenatal viral infection during pregnancy may affect the placenta and subsequent maternal and fetal health. Infectious exposure triggers maternal immune activation above physiological levels which can lead to exaggerated systemic immune response during gestation [24], and contribute to a pro-inflammatory placental environment [25]. This environment may potentiate adverse pregnancy outcomes such as stillbirth, preterm birth and permanent neurological defects [26]. Common placental pathologies associated with antenatal viral infection include vascular inflammatory lesions, villous fibrosis and altered villous maturation [27]. These lesions may arise from various inflammatory or immune processes affecting trophoblast function that alter placental development [28].

Antenatal infection co-occurring with maternal obesity may contribute to a “double burden” of disease [29,30], leading to poorer outcomes for mothers and offspring. While the mechanisms behind poor fetoplacental outcomes in pregnancies with obesity and viral infections are incompletely understood, antenatal infection could potentiate the inflammatory environment already present in obesity and could further exacerbate placental maldevelopment and dysfunction. Given the current global COVID-19 pandemic and the prevalence of maternal obesity, more information is needed regarding any potential combined effects of SARS-CoV-2 infection and obesity on placental development, fetal growth and pregnancy outcomes. While recent systematic reviews have linked SARS-CoV-2 infection, placental pathology and adverse pregnancy outcomes, the impact of maternal obesity on outcomes in the context of SARS-CoV-2 infection still needs to be examined. As a first step to fill this gap in knowledge, we conducted a systematic review and narrative synthesis of the current literature related to SARS-CoV-2 infection in pregnancy and maternal obesity. We hypothesized that high maternal body mass index (BMI) and SARS-CoV-2 infection leads to increased placental pathology and abnormal fetal growth to a greater degree than either of these exposures occurring alone. Our primary objective was to determine whether, and to what extent, elevated maternal BMI and SARS-CoV-2 infection influence placental phenotype. Our secondary objective was to investigate fetal outcomes and neonatal birth anthropometry in these pregnancies. With a better understanding of how obesity and SARS-CoV-2 infection during pregnancy impact the placenta, we may be able to identify pregnancies at greatest risk of adverse outcomes. This risk identification would allow for earlier interventions and monitoring to optimize maternal and fetal health. 

## 2. Methods

### 2.1. Reporting Guidelines

This study complied with the updated Preferred Reporting Items for Systematic reviews and Meta-Analysis (PRISMA) guidelines 2020 [31] and narrative synthesis guidelines [32].

### 2.2. Information Sources and Search Terms

The initial literature search was conducted between 1 April and 16 April 2021 using PubMed, CINAHL and ProQuest. The following pre-defined search terms were used: (pregnancy OR pregnant OR prenatal) AND (obes* OR BMI) AND (virus OR viral infection OR COVID-19 OR SARS-CoV-2) AND (placenta AND weight) AND/OR (placenta AND pathology) AND (birth AND size) AND/OR (birth AND weight) and yielded 5403 articles. After deduplication (EndNote v20.1.0.15043), 3157 articles remained. A further eight articles were removed following manual deduplication, leaving 3149 articles for screening. Because of the rapid output of literature related to the COVID-19 pandemic, a second and third search were conducted using the Corona Central database (https://coronacentral.ai/; accessed on 20 October 2021), where all COVID-19-related articles are indexed. The searches were conducted between 22 May and 31 May 2021, and between 7 October and 20 October 2021. Search terms used in Corona Central were (placenta), yielding 221 additional articles (May search; Figure 1C) and 316 articles (October search; Figure 1D). After deduplication with previously identified articles, 208 articles and 116 articles remained for screening. A final search on PubMed, CINAHL and ProQuest was conducted between 1 February and 15 February 2022 using the search terms described above in order to capture any studies on viral infection and elevated maternal BMI, in the publication date range that overlapped our searches of Corona Central (May and October 2021) which refers to the period Corona Central was searched). In this last search, 543 additional articles were captured. In total, 4029 articles were captured for screening across four distinct search periods. 

### 2.3. Inclusion Criteria

Inclusion and exclusion criteria were the same for all searches. Eligible study designs included randomized control trials (RCTs), case series, controlled clinical trials, case–control studies, cross-sectional studies, and cohort studies that reported placental pathology and/or neonatal anthropometry in cases of maternal exposure to any viral infection at any gestational age during pregnancy and high maternal BMI (BMI ≥ 25.0 kg/m^2^ before conception and/or measured during pregnancy) [33]. Inclusion criteria were (1) English language studies, (2) publication date between January 2010 and February 2022, (3) human studies, and (4) access to the full-text article. All articles that did not meet the inclusion criteria were excluded. There were no additional exclusion criteria. 

### 2.4. Article Screening and Data Collection

A 3-level screening process was performed by two reviewers for each of the four article searches described above (level 1 = title and abstract screening, level 2 = full-text screening, level 3 = data collection from eligible articles). In search 1 (Figure 1A), we targeted PubMed, CINAHL and ProQuest and captured 3149 articles (after deduplication). These articles were screened at level 1 and were excluded if they did not meet the inclusion criteria. After the first-level screening, 2933 articles were excluded, leaving a total of 216 articles for level 2 screening. At level 2, full-text articles were screened, and excluded if the study did not report on our outcomes of interest. Articles that only reported neonatal anthropometry were retained for analysis of secondary objectives. At the end of level 2 screening, the following studies were carried forward for screening at level 3: four studies reporting high maternal BMI, SARS-CoV-2 infection and placental pathology, and three studies reporting high maternal BMI, SARS-CoV-2 infection and neonatal anthropometry, but not placental pathology. After level 3 screening, these seven articles met final inclusion eligibility. However, placental pathology (*n* = 1) and neonatal anthropometry data (*n* = 5) could not be interpreted in 6/7 studies because they were not reported as a distinct category/group, leaving one study eligible for inclusion from search 1. 

Subsequent searches (*n* = 3) followed the same approach as described above. In brief, we searched PubMed, CINAHL and ProQuest again in February of 2022 (Figure 1B). We captured 543 articles for level 1 screening. Of these, 541 articles were excluded for not including elevated maternal BMI and viral infection during pregnancy, leaving two articles to be screened at level 2. At the end of level 2, these two articles met final inclusion eligibility for secondary objectives only, and data were extracted at level 3.

We also targeted the Corona Central database to capture studies on SARS-CoV-2 infection during pregnancy that reported BMI and our outcomes of interest (Figure 1C). In the first search of Corona Central (May of 2021), 221 articles were captured for level 1 screening. Of these, 13 articles were excluded, leaving 208 articles for level 2 screening. Following level 2 screening, 18 studies remained for level 3 where data were extracted. Of these, 14 articles were excluded as placental pathology and/or neonatal anthropometry data were not reported by maternal BMI. The other four studies reported placental pathology and birth anthropometry in pregnancies with maternal BMI ≥ 25.0 kg/m^2^ and SARS-CoV-2 infection (Figure 1 and Appendix A). These eligible articles were carried forward for data collection at level 3 and included in this review. We searched Corona Central a second time (October 2021; Figure 1D). In this search, 316 articles were captured and 116 articles remained after exclusion criteria were applied. At level 1 screening, 97 articles were excluded for not reporting maternal elevated BMI and viral infection during pregnancy, leaving 19 articles for level 2 screening. Following level 2 screening, 10 articles were excluded as they did not report placental pathologies and/or neonatal anthropometry. At level 3, data variables were extracted, as mentioned above, from the nine articles remaining. All SARS-CoV-2 studies were published between 2020 and 2022. 

Collectively across all searches, 16 articles were eligible for analysis. All but three articles were investigations related to SARS-CoV-2 in pregnancy. These three articles investigated HIV infection at the time of pregnancy. Given that these HIV articles did not report on placental pathologies, and only included newborn outcomes, we elected to exclude these articles and focus the present review on BMI and SARS-CoV-2 in pregnancy. In total, after this last exclusion, 13 articles were included in this review. 

### 2.5. Defining Outcomes

Our primary outcome of interest was the presence of placental pathologies in SARS-CoV-2 infected pregnancies stratified by maternal BMI category compared to non-infected pregnancies by maternal BMI category. Placental pathologies (histological lesions) were classified into ten broad etiological categories: evidence of maternal vascular malperfusion; evidence of maternal decidual arteriopathy; implantation site abnormalities; evidence of ascending intrauterine infection; evidence of placenta villous maldevelopment; evidence of fetal vascular malperfusion; evidence of utero-placental separation; fibrinoid; intervillous thrombi; and evidence of chronic inflammation [34,35]. All placental pathologies were described separately (Box 1). For case series studies, placental pathologies were reported stratified by maternal BMI (normal weight [NW, BMI 18.5–24.9 kg/m^2^; overweight (OW, BMI 25.0–29.9 kg/m^2^) and obese (OB, BMI ≥ 30.0 kg/m^2^). For cohort studies where the average maternal BMI was 25.0 kg/m^2^ and above, the presence and absence placental pathologies were further stratified by maternal SARS-CoV-2 infection status (SARS-CoV-2 positive and negative) during pregnancy. Based on population placental weight ratio distributions by sex [36], placental weight ≤ 10th centile was considered small for gestational age (SGA) and placental weight ≥ 90th centile was considered as LGA. Additionally, infant birthweight ≤ 10th centile for gestational age and sex was considered a proxy for FGR [37].

Box 1Placental pathology definitions and clinical significance.
**
Placental chronic inflammation
**
**Definition**: inflammatory response marked by the infiltration of lymphocytes, plasma cells, and histiocytes in the placental tissue and fetal membranes [38]. This inflammatory process can be identified in the villous tree, extraplacental membranes, chorionic plate, and basal plate of the placenta. Viral, bacterial and parasitic infections have been implicated in the etiology of placental chronic inflammation.**Clinical significance**: villitis of unknown etiology is associated with preterm and term FGR [39], SGA associated with preeclampsia, fetal death and chronic chorioamnionitis.**Pathologies included in this study classified as chronic inflammation**: acute/chronic villitis; acute/chronic intervillitis; acute/chronic deciduitis; chronic histiocytic intervillositis; histiocytic neutrophilic intervillositis; intervillous inflammation; basal villitis of unknown etiology.
**
Maternal vascular malperfusion (MVM)
**
**Definition**: group of placental lesions associated with abnormal uteroplacental blood flow [35].**Clinical significance**: MVM lesions occur in ~50% ofpreterm pregnancies [40]. MVM is associated with increased rates of fetal demise especially when extensive placental infarctions are present [41].**Pathologies included in this study classified as MVM**: villous infarction; villous agglutination; increased syncytial knots; accelerated villous maturation; distal villous hypoplasia.
**
Fetal vascular malperfusion
**
**Definition**: placental lesions associated with reduced or absent perfusion of the villous parenchyma by the fetus [42].**Clinical significance**: lesions such as fetal thrombosis and large foci of ischemic villi are associated with fetal demise, FGR, neonatal coagulopathies (thrombocytopenia, thromboembolic disease), and fetal cardiac anomalies [43,44]. Small foci of ischemic villi are associated with fetal encephalopathy [45].**Pathologies included in this study classified as FVM**: subchorionic thrombosis; high grade FVM; thrombi in the fetal circulation; extensive avascular villi; karyorrhexis.
**
Fibrinoid
**
**Definition**: extracellular material such as fibrin with other molecules derived from blood clotting or degenerative processes [46] accumulated in the placental villous, perivillous and intervillous space.**Clinical significance**: massive perivillous fibrin deposition (MPFD), which represents ≥25% of villi encased by fibrin, is associated with FGR and perinatal mortality [47]. MPFD is associated with recurrence risk in future pregnancies [48,49].**Pathologies included in this study classified as fibrinoid**: villous/intervillous fibrin; extensive perivillous and intervillous fibrin deposition; increased fibrin deposition; mild fibrin deposition.
**
Calcifications
**
**Definition**: calcium deposits in the placenta [50].**Clinical significance**: placental calcification has been associated with preeclampsia and FGR [51] and microcalcifications in placental villi could lead to oxidative stress [52].**Pathologies included in this study classified as calcifications**: calcifications; increased calcification.
**
Villous maldevelopment
**
**Definition**: evidence of delayed or accelerated placental villous development and/or maturation, which deviates from normal maturation for gestational age [53,54].**Clinical significance**: delayed villous maturation is associated with maternal metabolic disorder and obesity, intrauterine hypoxia, FGR, and fetal death [55].**Pathologies included in this study classified as villous maldevelopment**: chorangiosis; delayed villous maturation.
**
Hematoma
**
**Definition**: hemorrhagic lesions in the placenta [56].**Clinical significance**: cases of rounded intraplacental hematoma are associated with higher risk of decidual vasculopathy and infarction than other thrombohematomas and fetal demise [56].**Pathologies included in this study classified as hematoma**: hematoma.
**
Evidence of decidual vasculopathy
**
**Definition**: pathology of the maternal spiral arteries [57].**Clinical significance**: decidual vasculopathy is common in preeclampsia and other outcomes associated with preeclampsia such as FGR, placental abruption and fetal death [57].**Pathologies included in this study classified as decidual vasculopathy**: fibrinoid necrosis; decidual vasculopathy.
**
Intervillous thrombi
**
**Definition**: pathology consistent with leakage of fetal blood in the intervillous space and further blood clotting [58].**Clinical significance**: studies suggest an association of intervillous thrombi with gestational diabetes mellitus [59] and incidence of preeclampsia [60], FGR [61] and male sex [62].**Pathologies included in this study classified as intervillous thrombi**: intervillous thrombosis.
**
Villous edema
**
**Definition**: accumulation of fluid in the stroma of the chorionic villi, posing a barrier to gas exchange between mother and fetus [63].**Clinical significance**: this pathology is related to antenatal hypoxia [63].**Pathologies included in this study classified as villous edema**: diffuse villous edema and mature chorionic villi with focal villous edema.
**
Syncytiotrophoblast and trophoblast necrosis
**
**Definition**: syncytiotrophoblast and trophoblast death resulting in lysis of the cell which could lead to further tissue damage [64].**Clinical significance**: this pathology has been recently associated with SARS-CoV-2 placentitis [65].**Pathologies included in this study classified as syncytiotrophoblast and trophoblast necrosis**: syncytiotrophoblast and trophoblast necrosis.
**
Ischemia
**
**Definition**: the reduction of blood flow, and consequently oxygen, in the placenta [66].**Clinical significance:** placental ischemia is considered to be a key factor for preeclampsia [66] and common in women with pre-existing vascular disorders [67].**Pathologies included in this study classified as ischemia**: ischemia.
**
Erythroblast nuclear debris
**
**Definition**: findings of erythroblast nuclear remains, an intermediate cell in the initial stage of red blood cell formation generated in the yolk sac [68].**Clinical significance**: during embryonic development, the hematopoietic system supports embryo growth and survival through the differentiation of blood cells and a pool of hematopoietic stem cells [69].**Pathologies included in this study classified as erythroblast nuclear debris**: erythroblast nuclear debris.
**
Implantation site abnormalities
**
**Definition**: pathology consistent with various disorders that have decreased or absent decidualized endometrium, which could lead to abnormal placental implantation and increased placental adhesion [70] and invasive placenta [71].**Clinical significance**: this pathology is associated with maternal and fetal morbidity and mortality [70].**Pathologies included in this study classified as implantation site abnormalities**: basal plate myometrial fibres (stage 1); placenta accreta.
**
Macroscopic findings
**
**Definition**: macroscopic examination of the placenta includes general characteristics such as odour, color, shape, membranes (completeness, membrane rupture site measure), weight, fetal surface (color and appearance, surface and subchorionic region, fetal surface vessels), umbilical cord (length and diameter, spiralling, insertion, knots and umbilical vessels) and placental disk measurements [71]. **Pathologies included in this study classified as macroscopic findings**: Any macroscopic findings.

### 2.6. Defining Other Variables 

Additional variables of interest included trimester of pregnancy in which SARS-CoV-2 infection was diagnosed (stratified by first, second and third trimester); maternal SARS-CoV-2 symptomology (asymptomatic, mild, moderate, severe and critical, reported by the authors in each study); maternal co-morbidities (maternal pre-existing/gestational diabetes, hypertensive disorders of pregnancy including preeclampsia); and COVID-19 stringency index, a combined measure of nine different metrics of government responses to the COVID-19 pandemic [72]. In order to calculate the stringency index, nine metrics were used which were based on school closures, workplace closures, cancellation of public events, restrictions on public gatherings, closures of public transport, stay-at-home requirements, public information campaigns, restrictions on internal movements, and international travel controls. The COVID-19 stringency index is represented in a scale from 1 to 100. 

### 2.7. Grouping Studies for Synthesis

Of the 13 studies included for this review, seven were case series studies and six were cohort studies. All case series studies reported maternal infection with SARS-CoV-2, and we further categorized these SARS-CoV-2 positive cases by maternal BMI category. Some studies only reported maternal BMI ≥ 30 kg/m^2^. BMI below this cut-off was grouped as NW + OW (Figure 2). All six cohort studies reported an average maternal BMI ≥ 25.0 kg/m^2^, and data were further stratified by maternal diagnosis of SARS-CoV-2 (SARS-CoV-2 positive(+) or negative(−)) during pregnancy. For these six cohort studies, individual- or group-level BMI information was not reported; therefore, BMI stratification was not possible (Figure 2). Four cohort studies reported placental pathology and fetal anthropometry. Two studies only reported fetal anthropometry and were kept for secondary analysis. Further, one cohort study by Brien et al. included cohorts from three different countries (Canada, United Kingdom and France). The Canadian cohort included a historic control group of placentae from non-infected pregnancies by maternal BMI category collected prior to the SARS-CoV-2 pandemic (2016 to 2019) and compared this group to placentae from SARS-CoV-2(+) pregnancies (Figure 2). Since the cohort from France did not report an average of elevated maternal BMI, this population was not included. 

### 2.8. Data Synthesis and Visualisation

A Graphical Overview for Evidence Reviews (GOfER) diagram was used to synthesize key parameters of each eligible study including study design, maternal BMI (prepregnancy or measured during gestation), infection status (SARS-CoV-2(+)/(−)), gestational age at infection diagnosis, severity of SARS-CoV-2 symptoms (defined by asymptomatic, mild, moderate, severe and critical), prevalence of each placental pathology and newborn/birth outcome, and COVID-19 stringency index. We presented the lowest and highest values of the COVID-19 stringency index during each study period, based on the first day value of each month [72]. The mean stringency index was also calculated using the COVID-19 stringency index value for the first day of the month. Further, vaccination availability in each country at the time of the study was assessed and for all studies, vaccination status was not reported or COVID-19 vaccines were not available at the time of the patient’s recruitment. The prevalence of placental pathologies, stratified by maternal BMI categories and SARS-CoV-2 infection status, was visualized using heatmaps and bubble plots (JMP Software Version 16.1). A comprehensive list of any maternal comorbidity reported in the included studies was also presented (Appendix A). Finally, a world heat map was created (Excel v2206) to visualize the geographical locations of each included study. 

### 2.9. Study Quality Assessment 

Study quality appraisal was assessed for case series using the Quality Appraisal Tool for Case Series (18-item checklist) [73]. Detection of bias was assessed for cohort studies using the Newcastle–Ottawa Quality Assessment Scale for cohort studies [74].

### 2.10. Narrative Synthesis

We synthesized the data captured and reported these using the narrative Synthesis Without Meta-analysis (SWiM) guideline [32], as available data and groups were not sufficient to perform a meta-analysis. In this review, the majority of the articles included were case series (*n* = 7) and with limited BMI information and control groups (e.g., normal weight groups or SARS-CoV-2 negative comparators were not reported). Further, cohort studies (*n* = 6) lacked individual- or group-level BMI information, limiting group comparisons.

## 3. Results

### 3.1. Study Design, Location and Demographics 

The 13 included studies were carried out in 13 countries (Appendix A). The majority of studies were located in the United States (7/13, 54%). One study included patients from five different countries, but the precise locations were not specified in the article [75]. Eleven of the 13 articles (78.5%) reported placental outcomes, eight articles (61.5%) reported both placental pathologies and birth anthropometry and three articles (21.5%) only reported on birth anthropometry. 

### 3.2. Assessment of Study Methodology Quality 

Among the seven case series included, four of the case studies were conducted at single institutions [65,76,77,78], while two case series included cases from multiple institutions [75,79]. In one case series, the recruitment center was not specified [80]. Further, one of the four studies did not clearly describe the eligibility criteria [77], suggestive of increased risk of bias (Figure 3). In the study by Watkins et al., statistical tests to assess differences in outcomes by infection status were not adequately described or were absent from the methods [65]. Instead, this study reported the placental pathology and birth anthropometry in narrative format [65]. Overall, each case series clearly reported their objectives, participant characteristics, outcomes measures and conclusions suggesting that criteria for quality were generally met with low risk of bias (Figure 3A). 

Among the six cohort studies included, the criteria for quality were largely met, demonstrating adequacy in the description of participant characteristics, cohort exposures and outcome assessment. However, the study by Adhikari et al. [81] did not include a group of SARS-CoV-2(−) pregnancies and only analyzed SARS-CoV-2(+) pregnancies. Further, this study (Adhikari et al.) and Patberg et al. failed to report other maternal risk factors such as diabetes and hypertension that could contribute to placental pathologies and altered neonatal anthropometry [81,82]. Additionally, these studies did not conduct blinded assessments of placental histopathology, indicating a potential risk of bias and influence on the research findings [81,82] (Figure 3B).

### 3.3. SARS-CoV-2 Infection and Elevated BMI Associate with Placental Pathologies 

Macroscopic and microscopic placental pathology findings reported in SARS-CoV-2 infected pregnancies with elevated maternal BMI reported in 11 of the included studies are presented in Figure 4, Figure 5 and Figure 6. Eight of 19 placental pathology lesions (42.1%) were common between the case series and cohort studies, 6/19 (31.5%) pathologies were specific to cohort studies and 5/19 (26.3%) pathologies were specific to case series. Eleven lesions did not fall into any of the etiological categories and were grouped together in a miscellaneous category (Figure 4, Figure 5 and Figure 6). These legions were classified as any lesions, hematoma, macroscopic lesions, microscopic lesions, structural defects, villous edema, ischemia, implantation abnormalities, erythroblast debris, syncytiotrophoblast and trophoblast necrosis.

In the case series studies (*n* = 7), the most frequent placental lesions reported in SARS-CoV-2(+) pregnancies with high maternal BMI were chronic inflammation (71.4%, 5/7 studies), fetal vascular malperfusion (71.4%, 5/7 studies), maternal vascular malperfusion (85.7%, 6/7 studies) and fibrinoids (100%, 7/7 studies) (Figure 5 and Figure 7A). Ritchmann et al. [76] and Menter et al. [77] included SARS-CoV-2(+) pregnancies with high maternal BMI (BMI ≥ 25.0; *n* = 10 pregnancies across both studies) and reported chronic inflammation in 70% of the placentae (*n* = 7/10) and fibrinoids in 60% of placentae (*n* = 6/10). In 50% of these placentae, chronic inflammation and fibrinoid lesions occurred concurrently. Maternal vascular malperfusion and fetal vascular malperfusion occurred in 30% of placentae (both *n* = 3/10) with both lesions occurring simultaneously in one placenta (Figure 5 and Figure 7A). Acute chorioamnionitis was also reported in 100% (5/5) of placentae from SARS-CoV-2(+) pregnancies with high BMI by Ricthmann et al. [76], and of these five chorioamnionitis cases, 80% (4/5) also reported chronic inflammation. In the two case series by Jang et al. and Fan et al. [78,79] (*n* = 19 across both studies), the prevalence of maternal vascular malperfusion, fetal vascular malperfusion and fibrinoids was 31.5% (6/19) in SARS-CoV-2(+) pregnancies with normal BMI while the prevalence of the same pathologies was 10.5% (2/19) in SARS-CoV-2(+) pregnancies with high BMI (Figure 5 and Figure 7A). Additionally, Fan et al. [78,79] observed that 37.5% (*n* = 3/8) of placentae from SARS-CoV-2(+) pregnancies with normal BMI showed both maternal vascular malperfusion and fibrinoid lesions. Finally, in three case series [65,80], maternal BMI was only reported for participants when BMI was ≥ 30 kg/m^2^. When BMI was below this cut-off, participants were categorized as one group including NW and OW pregnancies since maternal BMI was not specified. In the study by Hanna et al. [80], chronic inflammation was observed in a single case of SARS-CoV-2(+) infection with obesity (Figure 5 and Figure 7A). In another study by Watkins et al. [65], fibrinoids and chronic inflammation were observed in 28.5% of SARS-CoV-2(+) pregnancies with maternal BMI ≥ 30 kg/m^2^ (*n* = 2/7 pregnancies; *n* = 8 placentae due to one pregnancy with twins). Comparatively, fibrinoids and chronic inflammation occurred simultaneously in all placentae (*n* = 6/7) from SARS-CoV-2(+) pregnancies with maternal BMI < 30 kg/m^2^. Further, fetal and maternal vascular malperfusion occurred in 66% (4/6) and 33% (2/6) of these placentae, respectively. In two of these six placentae, maternal and fetal vascular malperfusion were reported concurrently (Figure 5 and Figure 7A). Lastly, Schwartz et al. [75] reported placental pathology in SARS-CoV-2(+) pregnancies with maternal BMI ≥ 30 kg/m^2^ (*n* = 2) or maternal BMI < 30 kg/m^2^ (*n* = 9). In SARS-CoV-2(+) pregnancies with maternal BMI ≥ 30 kg/m^2^, chronic inflammation was reported in both placentae with fibrinoids and maternal vascular malperfusion also observed in one of these placentae. In SARS-CoV-2(+) pregnancies with maternal BMI < 30 kg/m^2^, chronic inflammation was observed in all placentae (*n* = 9/9) and fibrinoid in 55.5% (5/9) of placentae. Maternal and fetal vascular malperfusion were found in 22% (2/9) and 11% (1/9) of placentae, respectively. Additionally, in this group, trophoblast necrosis occurred in 66% (6/9) of placentae and syncytiotrophoblast necrosis in 55% (5/9) of placentae, with both lesions occurring concomitantly in three placentae (Figure 5 and Figure 7A).

In the cohort studies (*n* = 4), three studies [82,83,84] reported higher rates of chronic inflammation, maternal vascular malperfusion, fetal vascular malperfusion and fibrinoids in SARS-CoV-2(+) pregnancies with high maternal BMI (72%, *n* = 107/149; mean BMI of 30 kg/m^2^) compared to SARS-CoV-2(−) pregnancies with high BMI (7.4%, *n* = 10/135; Figure 6 and Figure 7B). In the fourth cohort study [81], common lesions observed in placentae from SARS-CoV-2(+) pregnancies with high BMI (*n* = 187 pregnancies; mean BMI of 30 kg/m^2^) were chronic inflammation (99%, 186/187), maternal vascular malperfusion (40%, *n* = 74/187) and fetal vascular malperfusion (26%, *n* = 48/187) (Figure 6 and Figure 7B). Of note, this particular cohort study did not include SARS-CoV-2(−) pregnancies as a control group.

Statistically significant differences in placental lesions between SARS-CoV-2(+) and SARS-CoV-2(−) pregnancies with high maternal BMI were reported by the authors in three out of the four cohort studies [82,83,84]. Brien et al. [83] reported a significant increase in maternal vascular malperfusion and excess fibrin deposition in SARS-CoV-2(+) pregnancies with mean BMI of 30 kg/m^2^, compared to non-infected pregnancies of the same BMI (maternal vascular malperfusion: 22.5% (7/31) vs. 0% (0/20), *p*-value < 0.05; fibrin deposition: 54.8% (17/31) vs. 25% (5/20), *p*-value < 0.05) (Figure 6 and Figure 7B). Additionally, the authors reported a significantly increased prevalence of accelerated villous maturation in SARS-CoV-2(+) pregnancies with mean BMI of 30 kg/m^2^ compared to the non-infected, high BMI group (16.1% (5/31) vs. 0% (0/38) *p*-value < 0.05; Figure 6 and Figure 7B). This study [83] also included a second cohort of infected and non-infected pregnancies and reported a significantly increased prevalence of placental excess fibrin deposition (*p*-value <0.05) and calcifications (*p*-value <0.001) in SARS-CoV-2(+) pregnancies with mean BMI of 30 kg/m^2^, compared to non-infected, high BMI pregnancies (Figure 6 and Figure 7B). Patberg et al. [82] reported a significant difference in fetal vascular malperfusion (*p*-value <0.001) and chronic inflammation (*p*-value <0.03) in SARS-CoV-2(+) pregnancies compared to SARS-CoV-2(−) pregnancies with high maternal BMI (Figure 6 and Figure 7B). Finally, Lu-Cullingan et al. [84] reported a significantly increased prevalence of placental fibrin deposition in SARS-CoV-2(+) pregnancies with mean BMI of 30 kg/m^2^ compared to SARS-CoV-2(−) pregnancies with mean BMI of 30 kg/m^2^ (33.3% (9/27) vs. 0% (0/10) pregnancies, *p*-value = 0.04 (Figure 6 and Figure 7B).

### 3.4. SARS-CoV-2 Infection and Elevated BMI Do Not Affect Placental Anthropometry

The majority of included studies did not report altered placental anthropometry in SARS-CoV-2(+) pregnancies with elevated BMI. Only two articles reported placental anthropometry. First, in Adhikari et al. [81], SGA placentae were reported in 50.8% (*n* = 31/61) of SARS-CoV-2(+) pregnancies with BMI ≥ 30 kg/m^2^ (Figure 6). Second, in the study conducted by Patberg et al. [82], 9% (*n* = 7/77) of placentae from SARS-CoV-2(+) pregnancies with mean maternal BMI of 30 kg/m^2^ were SGA, compared to 12.5% (*n* = 7/56) that were SGA in SARS-CoV-2(−) pregnancies with the same BMI (Figure 6). Further, this study also reported that 1.2% (*n* = 1/77) of placentae in SARS-CoV-2(+) pregnancies with mean BMI of 30 kg/m^2^ were LGA, compared to 1.7% (*n* = 1/56) of LGA placentae in SARS-CoV-2(−) pregnancies with mean BMI of 30 kg/m^2^ (Figure 6). The remaining studies did not report data on placental anthropometry. 

### 3.5. SARS-CoV-2 Infection and Elevated BMI Do Not Affect Birth Anthropometry 

High maternal BMI with SARS-CoV-2 infection does not appear to affect newborn anthropometry when compared to infected pregnancies with normal BMI or SARS-CoV-2(−) pregnancies with high BMI. Jang et al. [78] (case series study) reported FGR in one SARS-CoV-2(+) pregnancy where maternal BMI was ≤25.04.9 kg/m^2^ (1/7 pregnancies; Figure 5). In a cohort study by Adhikari et al. [81], 13% (*n* = 31/245) of newborns were SGA at birth in SARS-CoV-2(+) pregnancies with mean BMI of 30 kg/m^2^ compared to 10% (*n* = 316/3035) of newborns who were SGA in SARS-CoV-2(−) pregnancies with mean BMI of 30 kg/m^2^ (Figure 6). Brien et al. [83] (cohort study) reported a slight increase in preterm birth in a Canadian cohort of SARS-CoV-2(+) pregnancies with mean BMI of 30 kg/m^2^ (16.1%, *n* = 5/31) compared to SARS-CoV-2(−) pregnancies (7.9%, *n* = 3/38) and historic controls (0%, *n* = 0/20) with mean BMI of 30 kg/m^2^. No adverse birth outcomes were reported in the UK cohort. Further, consistent with the findings of generally unaltered birth anthropometry, no other birth complications were found, or they were not reported [65,75,76,77,79,80,82,84,85,86]. 

### 3.6. SARS-CoV-2 Infection and Elevated BMI Associate with Fetal Demise 

Importantly, some studies of SARS-CoV-2 infection during pregnancy reported intrauterine fetal demise. Forty-three percent (3/7) of case series of SARS-CoV-2(+) pregnancies with elevated maternal BMI reported fetal demise [75,76]. Specifically, in 57.8% (11/19) of cases in the cases series studies, fetal demise was reported in SARS-CoV-2(+) pregnancies. Of these 11 cases, 63.6% (7/11) were pregnancies where mothers had elevated BMI. In the study by Ritchmann et al. [76] (case series), fetal demise occurred in all pregnancies with high BMI (*n* = 5) across a range of gestational ages (21 to 38 weeks of gestation), and all deaths occurred within 22 days of SARS-CoV-2 diagnosis. Further, acute histological chorioamnionitis in the placenta and neutrophils in the alveolar spaces were observed in one fetus (demise occurring at 21 weeks’ gestation), suggestive of congenital SARS-CoV-2 infection. The study conducted by Schwartz et al. [83] reported 45.4% (5/11) fetal deaths in SARS-CoV-2(+) pregnancies occurring between 22 and 39 weeks of gestation and high maternal BMI was confirmed in 20% (*n* = 1/5) of the fetal death cases. Additionally, Schwartz et al. [75] described that 100% (6/6) of the live neonates tested positive for SARS-CoV-2 after birth, confirmed by qPCR test after birth. Finally, Hanna et al. [80] (case series of two SARS-CoV-2(+) pregnancies) reported 50% (1/2) of fetal death occurring at 36 weeks’ gestation in a pregnancy where maternal BMI was ≥ 30 kg/m^2^. Further, no fetal demises were reported in any of the cohort studies. 

## 4. Discussion

Here we used a systematic review and narrative synthesis to consolidate evidence on placental pathologies and adverse birth outcomes reported in human studies of maternal SARS-CoV-2 infection with elevated BMI during pregnancy. Our findings show that in SARS-CoV-2(+) pregnancies with elevated BMI, chronic inflammation, maternal vascular malperfusion, fetal vascular malperfusion and fibrinoids are commonly observed in the placenta and with greater prevalence when compared to SARS-CoV-2(−) pregnancies with elevated BMI. Further, increased fetal demise was reported in some case series of SARS-CoV-2(+) pregnancies with elevated BMI [75,76,80]. 

In this review, inflammatory lesions were commonly reported in placentae from SARS-CoV-2(+) pregnancies with high maternal BMI and frequently co-occurred with fibrinoid lesions. Chronic inflammation is associated with infiltration of lymphocytes, plasma cells, and/or macrophages in the placental tissue and can occur in the chorionic villi, fetal membranes and/or the decidua basalis [87] (Figure 8). Chronic inflammation in the placenta is associated with FGR, SGA associated with preeclampsia and fetal death, all serious complications of pregnancy that are known to arise due to placental dysfunction [88]. Further, studies demonstrate an association between elevated maternal BMI and chronic inflammatory lesions [88]. Systemically, increased adipose tissue accumulation, as seen in obesity, can lead to increased production of interleukin 6 (IL-6), interleukin IL-1β (IL-1β) and tumor necrosis factor (TNF-α) [89], pro-inflammatory cytokines able to disrupt the immune response at the maternal–fetal interface. During pregnancy, obesity is associated with increased levels of pro-inflammatory cytokines such as IL-6 in the placenta [90]. This pro-inflammatory intrauterine environment may result in altered trophoblast function, nutrient transport and/or inflammatory responses that contribute to placental dysfunction and adverse outcomes [91]. The additional insult of viral infection may further exacerbate this suboptimal environment and lead to increased placental damage and/or dysfunction. For example, the immune response may activate trophoblasts to secrete immunomodulators, such as antiviral interferons, that may have cytotoxic and pro-inflammatory effects on gestational tissue [92] and further increase the risk of placental maldevelopment (Figure 8). Further, a study in mice exposed to a viral mimic and high-fat diet during pregnancy demonstrated increased placental expression of inflammatory genes with both exposures compared to each occurring in isolation [93]. However, the exact mechanisms linking maternal obesity and antenatal infection to poor pregnancy outcomes are not yet known. Collectively, these findings highlight the need to understand how the co-occurrence of elevated maternal BMI and viral infection during pregnancy can impact the maternal–fetal interface leading to an imbalanced pro-inflammatory environment.

We also describe maternal vascular malperfusion in SARS-CoV-2(+) pregnancies with high maternal BMI. Prior research demonstrates that obesity is a risk factor for the occurrence of placental pathology, specifically maternal vascular malperfusion lesions [94]. How obesity could lead to placental vascular lesions is not completely understood, but some mechanisms have been proposed [95]. The inflammatory environment at the maternal–fetal interface in maternal obesity could lead to abnormal placentation including insufficient extravillous trophoblast (EVT) invasion and altered spiral artery remodeling (Figure 8). In turn, these abnormalities could affect placental vasculature development in the villi and impede placental nutrient and oxygen exchange [95]. Viral infections during pregnancy are also known to alter EVT invasion and lead to poor vasculature development [95]. Both exposures could collude to increase the risk of placental maldevelopment and poor fetal outcome. 

We also found increased prevalence of fibrin deposition and fetal vascular malperfusion in SARS-CoV-2(+) pregnancies with high maternal BMI, which are known to be associated with SGA, preterm delivery and fetal demise [42,46] (Figure 8). These lesions are reflective of altered vascular development in the placenta, altered uteroplacental blood flow and poor maternal–fetal exchange [96,97]. However, little is known about the mechanisms driving fibrin deposition and fetal vascular malperfusion, and the effects of these pathologies on gestational trajectory and fetal outcome in pregnancies where obesity and viral infections co-occur. More studies are needed to understand these relationships. 

According to the studies included in this review, birth anthropometry was not altered by either maternal infection with SARS-CoV-2 or elevated BMI during pregnancy. The only case of reported FGR was suggested to be the direct result of maternal vascular malperfusion [43]. Exposure to obesity or viral infection in isolation during pregnancy may have opposing effects on fetal growth. In humans, an increased risk for delivering an LGA neonate is seen in pregnancies with co-morbid obesity and diabetes [98]. Data on birthweight in pregnancies with elevated BMI are less consistent, and other studies have demonstrated SGA and appropriate for gestational age babies in pregnancies with obesity [98]. In contrast, studies of viral infections during pregnancy (e.g., Zika virus and Cytomegalovirus) demonstrate an increased risk of delivering low birth weight or SGA neonates [24,95,99]. Reduced fetal growth following infections during pregnancy may be due to the action of pro-inflammatory cytokines on EVT invasion [95], characteristic of maternal vascular malperfusion pathology, which associates with altered uterine and intervillous blood flow [95]. Other pathologies may impede fetal growth including fibrin deposition which is known to reduce villous volume and surface area [59]. Collectively, these placental pathologies could lead to altered maternal–fetal exchange and reflect on poor fetal growth. When obesity (particularly with hyperglycemia) and infections co-occur, it is possible that the opposing effects of these exposures on fetal growth may result in no signs of altered fetal anthropometry; however, future studies should test this hypothesis. 

Strengths of this review include rigorous study selection criteria, data interpretation based on both elevated maternal BMI and infectious exposure, and attempts to capture the most up-to-date peer-reviewed articles. However, there are some limitations to the quality of the findings of included studies and thus, generalizability of the results. While most of the studies included in this review had a low risk of bias, one study (case series) failed to clearly explain their inclusion criteria [77]. Moreover, four of the case series (4/7) [65,76,77,78] collected data from single hospital centres, potentially introducing selection bias and external validity. While Schwartz et al. included pregnancies from five different countries, the countries of origin were not provided, limiting our inability to assess the impact of geography. Additionally, the results for the case series were often reported in a narrative format, although relevant placental pathologies and birth outcomes were provided for each case. The inclusion of pregnancies with other maternal morbidities could also influence the presence and severity of placental pathologies observed in the included studies. Pregnancies with gestational diabetes and gestational hypertension were included [65,75,77,86] and could influence placental pathology and birth outcomes in pregnancies with SARS-CoV-2 infection, independent of maternal obesity. Finally, due to the temporal nature of the pandemic and mutations of the SARS-CoV-2 virus, we cannot ignore the potential for different viral strains affecting placental health and subsequent outcomes to varying degrees. However, SARS-CoV-2 strains were not identified in the included studies. Future studies should investigate how SARS-CoV-2 variants can impact gestational tissue development and fetal outcomes.

## 5. Conclusions

The findings of this systematic review suggest that maternal obesity and antenatal infection with SARS-CoV-2 influence placental health and together, could lead to worse outcomes for babies. Despite the novelty of this research in assembling major placental pathologies and adverse birth outcomes reported in human studies of maternal SARS-CoV-2 infection and elevated BMI during pregnancy, there is still limited information on how SARS-CoV-2 (and other viral infections) coupled with high maternal BMI can influence pregnancy trajectory and perinatal and postnatal outcomes. Here, we suggest future directions and research perspectives in the field of fetoplacental development and growth in pregnancies complicated by viral infection and overweight/obesity (Box 2).

Box 2Future directions and research perspectives in fetoplacental development in pregnancies complicated by viral infection and overweight and obesity.
Determine if placental pathologies and fetal growth differ based in placental sex in cases of maternal exposure to SARS-CoV-2 and elevated BMIDetermine in which trimester the placenta and fetus are most vulnerable to the effects of maternal infection with SARS-CoV-2 and overweight/obesity, and how timing of exposures influence:◦occurrence of placental pathologies◦vertical transmission◦offspring programmingDetermine if SARS-CoV-2 symptomology (mild to severe) and other conditions heightened during pandemics (e.g., maternal stress) differentially affect:◦placental pathologies◦maternal and fetal inflammation◦vertical transmissionDetermine the role of the maternal vaginal and gut microbiomes in pregnancies complicated by maternal SARS-CoV-2 and obesity and whether the microbiomes influence placental growth and development and consequently, fetal developmentDetermine if a treatment targeting the gut microbiome is able to reverse or prevent poor placental development and consequently, improve fetal growth via changes at the maternal–fetal interfaceDetermine if vaccines prevent adverse placental development and function and protect fetal growthDetermine if other maternal comorbidities influence:◦susceptibility to viral infection◦occurrence of placental pathology◦altered maternal and gestational tissue proinflammatory milieux◦increased vertical transmission◦long term offspring developmental programming 


Ultimately, there is a great need to consider extensive maternal health history that could underlie poor placental development and function and how multiple exposures interact to influence pregnancy and developmental trajectories. Such investigations could support development of targeted therapeutic and/or intervention strategies to reduce adverse outcomes.

## Figures and Tables

**Figure 1 pathogens-12-00524-f001:**
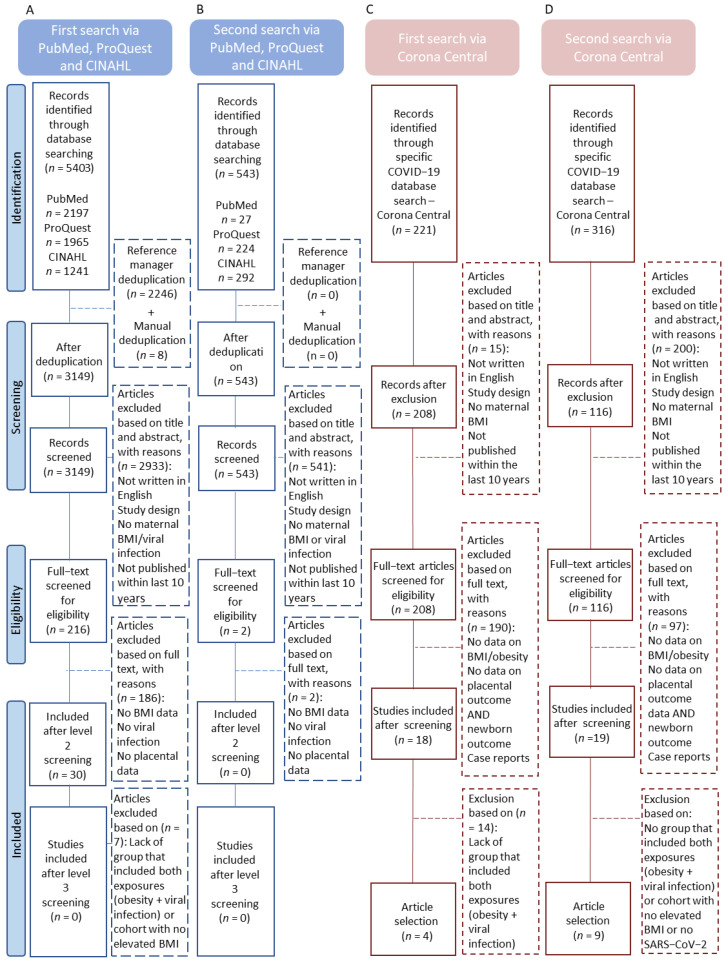
PRISMA flow diagram for article screening. (**A**) First article search using PubMed, ProQuest, and CINAHL databases conducted between 1 April and 16 April 2021. (**B**) Second article search using PubMed, ProQuest, and CINAHL databases conducted between 1 February 2022 and 15 February 2022. (**C**) First article search using the Corona Central database conducted between 22 May and 31 May 2021. (**D**) Second article search using the Corona Central database conducted between 7 October and 20 October 2021.

**Figure 2 pathogens-12-00524-f002:**
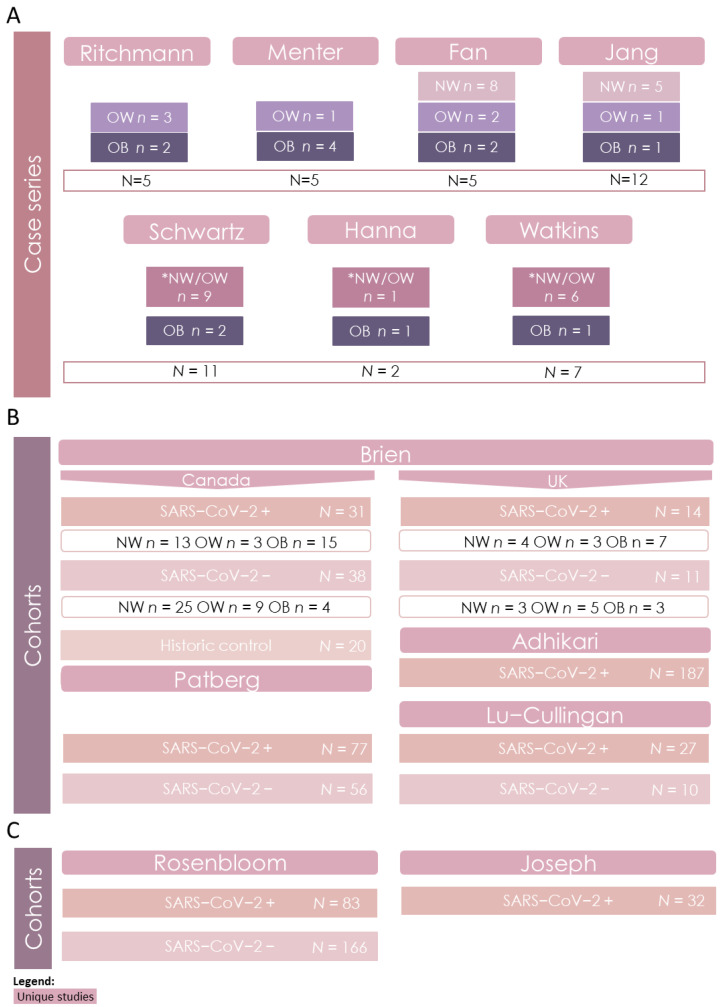
Grouping studies for evidence synthesis. (**A**) Grouping of case series studies (*n* = 7) for synthesis of placental pathology and birth anthropometry outcomes. SARS-CoV-2 (+) pregnancies were further stratified by maternal BMI. (**B**) Grouping of cohort studies (*n* = 4) for synthesis of placental pathology. Cohort studies where mothers had an average BMI of 25.0 kg/m^2^ and above were further stratified by maternal SARS-CoV-2 infectious status (SARS-CoV-2(+) and SARS-CoV-2(–). (**C**) Grouping of cohort studies (*n* = 2) for synthesis of birth anthropometry outcomes. *N* = the total number of cases in the study. *n* = the number of cases stratified by maternal BMI or SARS-CoV-2 status. * SARS-CoV-2(+) pregnancies where maternal BMI ≤ 30 kg/m^2^ was not specified.

**Figure 3 pathogens-12-00524-f003:**
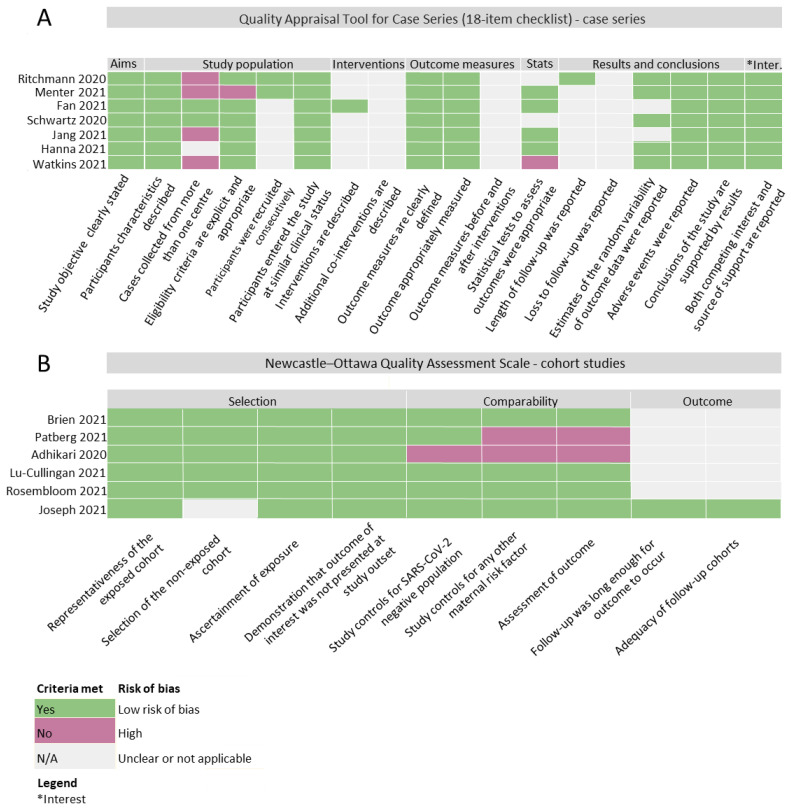
Quality assessment of articles according to study design. (**A**) Quality Appraisal Tool for Case Series (18-item checklist). (**B**) Newcastle–Ottawa Quality Assessment Scale (cohort studies).

**Figure 4 pathogens-12-00524-f004:**
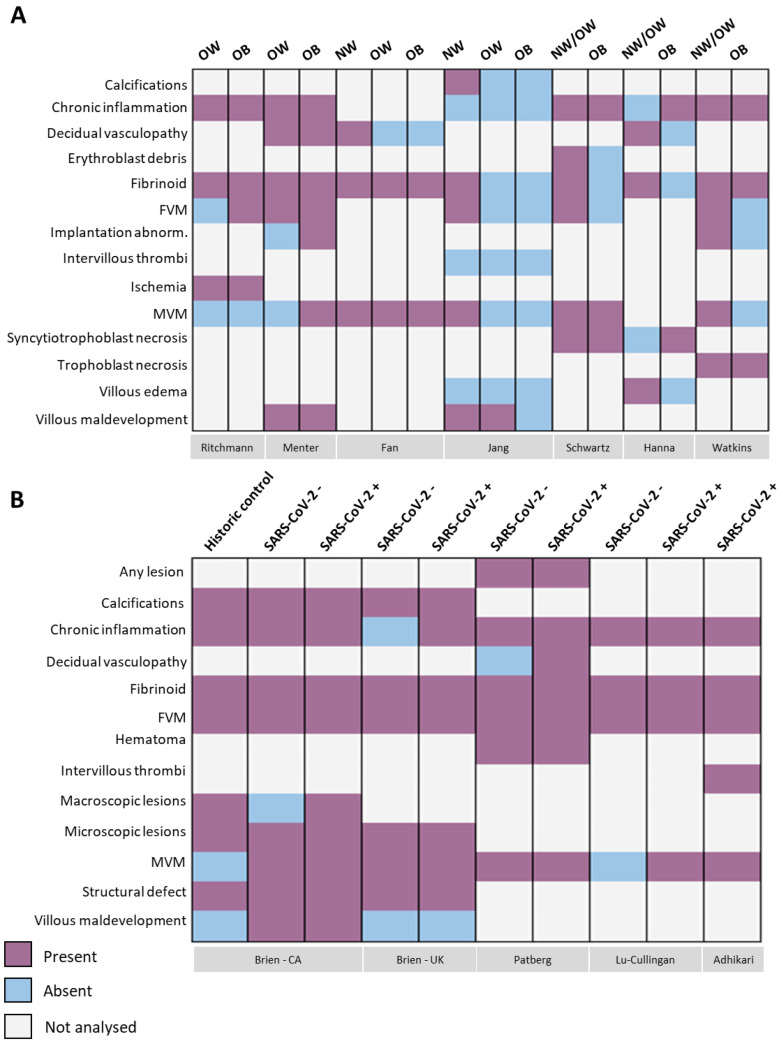
Heatmap of placental pathology presence. (**A**) Placental pathology presence in pregnancies with confirmed SARS-CoV-2 infection stratified by maternal BMI (case series). (**B**) Placental pathology presence in pregnancies with elevated BMI (mean BMI 30 kg/m^2^) stratified by SARS-CoV-2 infectious status (cohort studies). In one cohort study [83] placental pathologies from pregnancies with elevated BMI were analyzed between 2016 and 2019 and are considered by the authors as a historic control group. SARS-CoV-2− = SARS-CoV-2 negative; SARS-CoV-2+ = SARS-CoV-2 positive; FVM = fetal vascular malperfusion; MVM = maternal vascular malperfusion. Purple = placental pathology; blue = no placental pathology; grey = not analyzed.

**Figure 5 pathogens-12-00524-f005:**
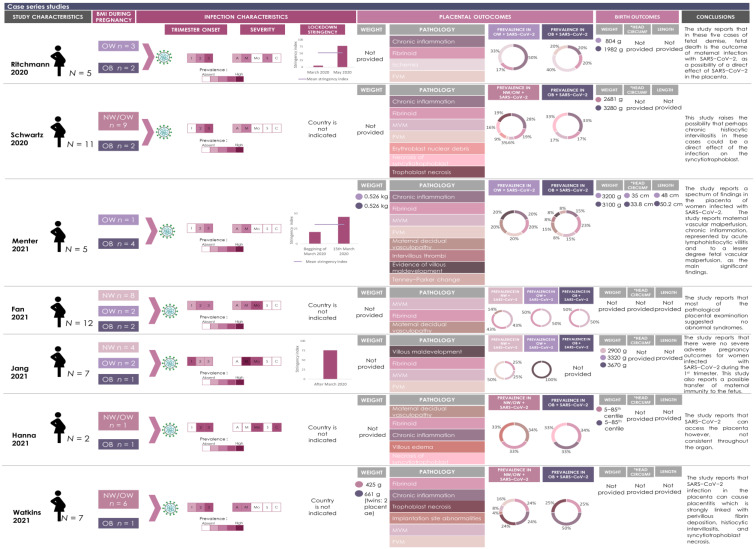
Graphical overview for evidence reviews (GOfER) diagram of case series reporting the main findings of placental pathologies and birth anthropometry in cases of maternal infection of SARS-CoV-2 and stratified by maternal BMI (OB—obesity/OW—overweight/NW—normal weight). Rectangles of different colors represent different maternal BMI categories associated with viral infection. 
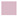
 represents NW women; 
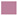
 represents women who are either NW or OW; 
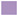
 represents OW women and 
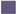
 represents OB women. The prevalence of viral infection during each trimester is represented by different shades of pink (higher the n, darker the color. Blank means absent). The severity of the infection is categorized by asymptomatic (A); mild (M); moderate (Mo), severe (S) and critical (C) and the prevalence is represented by different shades of pink (higher the n, darker the color. Blank means absent). The lockdown stringency in each country during the timeline of the study is represented as a bar chart. The mean lockdown stringency index is indicated for studies that were conducted over one month. Each placental pathology reported in the study was further categorized according to the synoptic framework for placental pathology. The prevalence of placental pathologies is represented in pie charts and categorized by maternal BMI. Rectangles of different colors are representing the placental pathologies and correspond with the colors of the placental pathologies represented in the pie chart. Placental pathologies in cases of SARS-CoV-2 infection are represented by the prevalence (*n* number) stratified by BMI category. Placental and birth anthropometry values are represented by the mean of each maternal BMI category. *Head Circumf = head circumference.

**Figure 6 pathogens-12-00524-f006:**
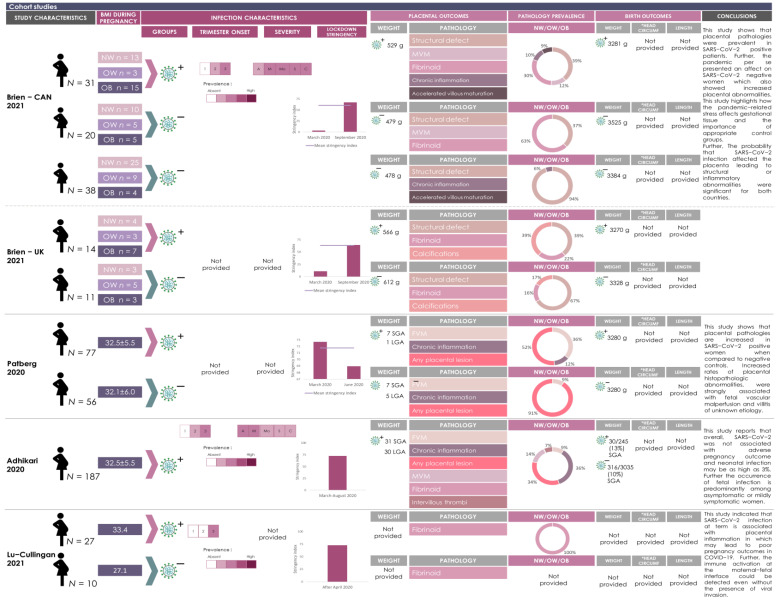
Graphical overview for evidence reviews (GOfER) diagram of cohort studies reporting the main findings of placental pathologies and birth anthropometry in cases of maternal infection of SARS-CoV-2. All cohorts included are within the obesity range. The prevalence of viral infection during each trimester is represented by different shades of pink (higher the *n*, darker the color. Blank means absent). The severity of the infection is categorized by asymptomatic (A); mild (M); moderate (Mo), severe (S) and critical (C) and the prevalence is represented by different shades of pink (higher the *n*, darker the color. Blank means absent). The lockdown stringency in each country during the timeline of the study is represented as a bar chart. The mean lockdown stringency index is indicated for studies that were conducted over one month. Placental pathologies represented in this GOfER were significantly different between SARS-CoV-2 positive and negative cohorts and further categorized according to the synoptic framework for placental pathology. The prevalence of placental pathologies is represented by infectious group (SARS-CoV-2 positive cohort (+) and negative cohort (−). The prevalence of placental pathologies in each cohort is represented in pie charts. Rectangles of different colors are representing the placental pathologies and correspond with the colors of the placental pathologies represented in the pie chart. Placental and birth anthropometry values are represented by the mean of each infectious group. *Head Circumf = head circumference.

**Figure 7 pathogens-12-00524-f007:**
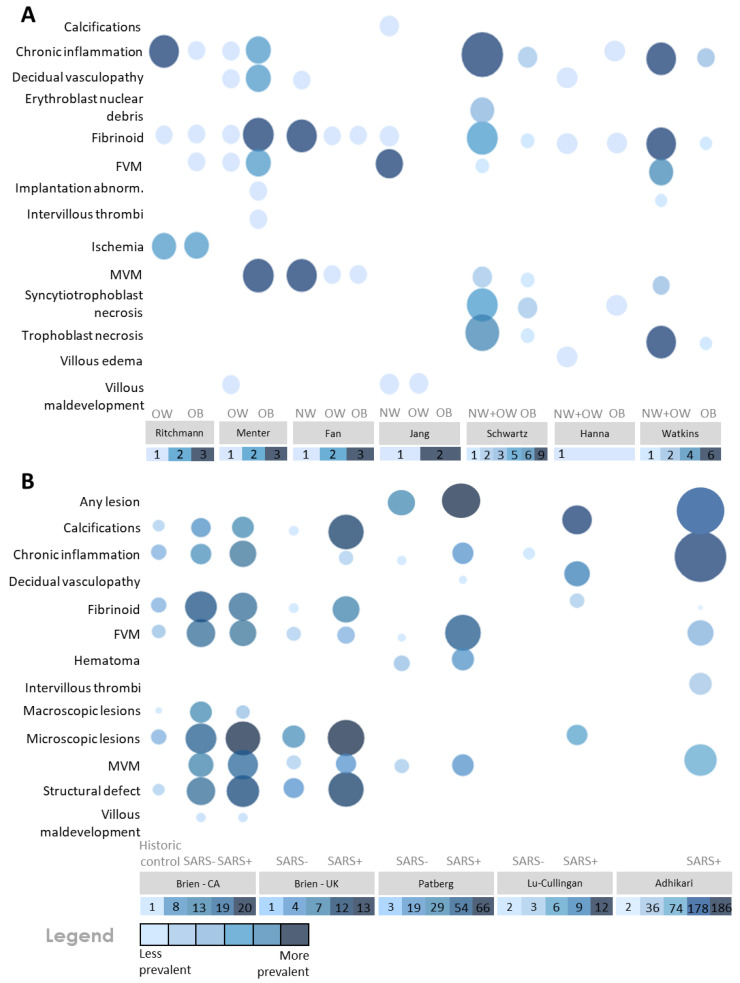
Bubble plot of prevalence of placental pathologies. (**A**) Prevalence of placental pathologies in pregnancies with confirmed SARS-CoV-2 infection stratified by maternal BMI (case series). (**B***)* Prevalence of placental pathologies in pregnancies with elevated BMI (mean BMI 30 kg/m^2^) stratified by SARS-CoV-2 infectious status (cohort studies). In one cohort study [83], placental pathologies from pregnancies with elevated BMI were analyzed between 2016 and 2019 and are considered by the authors as a historic control group. Higher prevalence of a given placental pathology is indicated by bubbles that are larger and darker in color. Lower prevalence of a given placental pathology is indicated by bubbles that are smaller and lighter in color. SARS-CoV-2− = SARS-CoV-2 negative; SARS-CoV-2+ = SARS-CoV-2 positive; FVM = fetal vascular malperfusion; MVM = maternal vascular malperfusion.

**Figure 8 pathogens-12-00524-f008:**
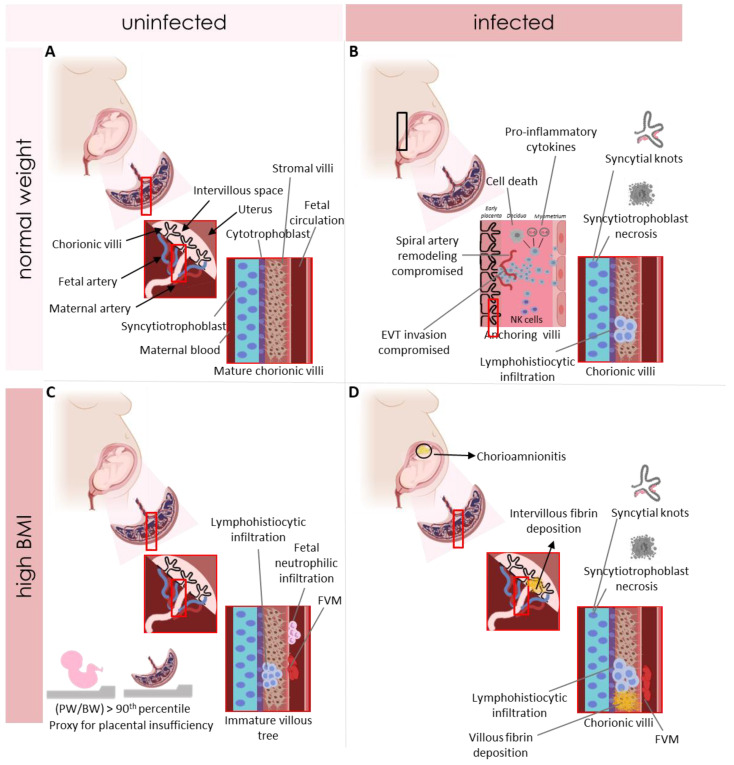
Placental pathologies by maternal BMI and viral infection status. (**A**) Placental pathologies in women with normal weight and uninfected during pregnancy—showing mature chorionic villi in normal weight and uninfected women. (**B**) Placental pathologies in women with normal weight and episodes of viral infection during pregnancy—showing early placental development in cases of viral infection with *CMV/HSV* leading to poor extravillous trophoblast invasion and inadequate spiral artery remodeling and increased pro-inflammatory cytokines such as IL-6 and IL-8. Additionally, representation of a chorionic villi with accelerated villous maturation (represented by increased syncytial knots), syncytiotrophoblast necrosis and villitis and intervillitis with lymphohistiocytc infiltration reported in cases of SARS-CoV-2. (**C**) Placental pathologies in women with elevated BMI during pregnancy and uninfected—illustrating an immature villous tree, chronic villitis, fetal vascular thrombosis (FVM) fetal neutrophilic infiltration and also increased placental and birth weight which is a proxy for placental insufficiency. (**D**) Summary of placental pathologies in SARS-CoV-2+ women with elevated BMI reported in this review—showing villitis and intervillitis, villous and intervillous fibrin deposition. Fetal vascular malperfusion (FVM), maternal vascular malperfusion (MVM) (represented by placental infarctions and accelerated villous maturation, represented by increased syncytial knots), syncytiotrophoblast necrosis predominantly in the areas of intervillositis and associated with massive perivillous fibrin deposition. Additionally, chorioamnionitis in histological samples was extensively reported across the studies. EVT = extravillous trophoblast invasion. FVM = fetal vascular malperfusion. PW = placental weight. BW = body weight. SARS-CoV-2+ = SARS-CoV-2 positive.

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
