# Peer review of "Impact of Co-Occurrence of Obesity and SARS-CoV-2 Infection during Pregnancy on Placental Pathologies and Adverse Birth Outcomes: A Systematic Review and Narrative Synthesis"

_pathogens, 2023, doi:10.3390/pathogens12040524_

Round 1
Reviewer 1 Report
In this manuscript, the authors systematically reviewed data on obesity- and SARS-CoV-2-related placental pathologies and adverse pregnancy complications. It was well-organized and nicely written. The descriptions were clear and informative. The tables and figures were informative, although the size of the font was too small to clearly be seen. The reviewer had several minor concerns:
1. As a review paper, the methods should be concise. Callout Box 1 should be moved to Supplementary materials.
2. In Figures 1-3, 5, 6, and 8, the size of the font was too small.
Reviewer 2 Report
The title doesn't make sense and could be change to Impact of co-occurrrence of obesity and SARS-CoV-2 infection during pregnancy on placental pathologies and adverse birth outcomes: a systematic review and narrative synthesis
Abstract and later in the text: It is not clear why fibrinoids (1%) are included as common lesions when other pathologies are being repoted at much higher rates 26% - 99%.
Section 2.3 If there were no exclusion criteria state this. At the moment only inclusion criteria are mentioned. The search period starts January 2010 - which seems bizarre when SARS-CoV-2 has to be mentioned as an inclusion critirion. Would be good to know the date (and study period) of the first eligible paper.
Grammar - ranging from Jan 2010 to (not and) Oct 2021 Section 2.4 is overly long. Suggest start with a summary of the level 1,2 and 3 screening of all papers. This really is all the reader needs. The detail about each search could be part of supplementary data along with Figure 1 which can be replaced by a simpler figure.
Call Box 1 - delete the second 'is' after VUE; space between maturation and is in Villous Maldevelopment Figure 2A - explain the numbers in the clear boxes Figure 4 - lowest row in and A and B is too small to read Figure 5 (and tex) - since the liklihood of vaccination is "not vaccinated" for each study - why not simply state this.
Figure 5 and 6 - too small to be useful - These need to be redrawn to be useful. What was the association between Covid stringency (sometimes referred to as lockdown stringency) and outcome?
Results 3.3 - Please check that the greater than > and less than < signs have been correctly used. The conclusion is that Obesity is associated with pathology but this is not how I read the results.
3.4 - re-state that the data presented here are from 2 studies only
3.6 - state how the cases were included and confirm that in all the pregnancies with high BMI the outcome was fetal demise (Ritchmann). Can a percentage outcome be safely used with these studies?
All live neonates "tested positive" for SARS-CoV-2 - specificy the testing used - presumably PCR?
4 Discussion - Figure 8 does not illustrate the statement of this sentence. There is another place to refer to the Figure which illustrates pathological pathways?
4/7 case series were from 1 centre - please confirm that case duplication was excluded.
